# Body mass index and all-cause mortality in HUNT and UK biobank studies: revised non-linear Mendelian randomisation analyses

Stephen Burgess ![ORCID],[1,2,3] Yi-Qian Sun ![ORCID],[4,5,6] Ang Zhou,[1,7] Christopher Buck,[1] Amy M Mason,[2,3] Xiao-Mei Mai ![ORCID] [8]

For numbered affiliations see end of article.

**Correspondence to**
Stephen Burgess;
sb452@medschl.cam.ac.uk

## ABSTRACT

**Objectives** To estimate the shape of the causal relationship between body mass index (BMI) and mortality risk in a Mendelian randomisation framework.

**Design** Mendelian randomisation analyses of two prospective population-based cohorts.

**Setting** Individuals of European ancestries living in Norway or the UK.

**Participants** 56 150 participants from the Trøndelag Health Study (HUNT) in Norway and 366 385 participants from UK Biobank recruited by postal invitation.

**Outcomes** All-cause mortality and cause-specific mortality (cardiovascular, cancer, non-cardiovascular non-cancer).

**Results** A previously published non-linear Mendelian randomisation analysis of these data using the residual stratification method suggested a J-shaped association between genetically predicted BMI and mortality outcomes with the lowest mortality risk at a BMI of around 25 kg/ m². However, the 'constant genetic effect' assumption required by this method is violated. The reanalysis of these data using the more reliable doubly-ranked stratification method provided some indication of a J-shaped relationship, but with much less certainty as there was less precision in estimates at the lower end of the BMI distribution. Evidence for a harmful effect of reducing BMI at low BMI levels was only present in some analyses, and where present, only below 20 kg/m². A harmful effect of increasing BMI for all-cause mortality was evident above 25 kg/m², for cardiovascular mortality above 24 kg/m², for cancer mortality above 30 kg/m² and for non-cardiovascular non-cancer mortality above 26 kg/m². In UK Biobank, the association between genetically predicted BMI and mortality at high BMI levels was stronger in women than in men.

**Conclusion** This research challenges findings from previous conventional observational epidemiology and Mendelian randomisation investigations that the lowest level of mortality risk is at a BMI level of around 25 kg/ m². Our results provide some evidence that reductions in BMI will increase mortality risk for a small proportion of the population, and clear evidence that increases in BMI will increase mortality risk for those with BMI above 25 kg/m².

---

## STRENGTHS AND LIMITATIONS OF THE STUDY

⇒ Mendelian randomisation design minimises bias due to confounding and reverse causation.
⇒ Large sample sizes enable powerful analyses even in low body mass index (BMI) individuals.
⇒ Validity of the genetic variants as instrumental variables cannot be verified.
⇒ Bias due to selection could be non-negligible and could vary across strata.
⇒ All estimates are averaged across a stratum of the population; individual effects of raising or lowering BMI may vary between individuals.

## INTRODUCTION

Body mass index (BMI) is a convenient and accessible measure of obesity. The epidemiological relationship between BMI and mortality is complex, with many observational studies conducted in Western countries showing a J-shaped relationship between BMI and mortality risk in the general population, such that mortality risk is lowest for those in the upper normal weight (BMI 20.0–24.9 kg/m²)[1–3] or even the overweight (BMI 25.0–29.9 kg/m²)[4–6] category rather than the lower normal weight (BMI 18.5–19.9 kg/ m²) or underweight category (BMI<18.5 kg/ m²). However, such findings may not reflect the causal relationship between BMI and mortality, as observational associations are influenced by confounding and reverse causation.

Mendelian randomisation is an epidemiological technique for the analysis of observational data designed to avoid bias due to confounding and reverse causation.[7] Rather than assessing associations between BMI and mortality directly, we assess associations between genetic variants that predict BMI levels and mortality risk.[8] This is analogous to the analysis of a randomised controlled trial for weight loss,[9] which would assess

associations between the randomised intervention and the outcome, rather than associations with measured BMI in trial participants. According to Mendel's laws of inheritance, genetic variants should be uncorrelated with traits that they do not affect, and hence should be independent of potential confounders. As genetic variants are determined at conception, genetic associations should be protected from reverse causation.[10] These properties mean that genetic variants are plausible instrumental variables; an instrumental variable is a variable that behaves as if it has been randomised in the population.[11] Hence, associations between genetic predictors of BMI and mortality provide insights into the causal effect of BMI on mortality.

Non-linear Mendelian randomisation is an extension of standard Mendelian randomisation to investigate the shape of the causal relationship between an exposure and an outcome.[12] A typical randomised trial estimates an average causal effect, representing the average impact of a population-wide shift in the distribution of an exposure; similarly, a standard Mendelian randomisation analysis estimates a population-averaged causal effect.[13] A widely used method for non-linear Mendelian randomisation stratifies the population based on levels of the exposure, and estimates 'localised average causal effects', representing the average impact of a population-wide shift in the distribution of the exposure for that stratum of the population.[13]

However, stratifying the population such that the instrumental variable assumptions still hold in the strata of the population is tricky. Stratifying on the exposure directly would break randomisation, as exposure levels are influenced by the instrumental variables. Hence, an individual in the lowest stratum of the population defined by the exposure with a genetic predisposition to high values of the exposure would be more likely to have low values of the confounders. This is an example of collider bias:[14] the exposure is a common effect of the instrument and exposure—outcome confounders, and so stratification on the exposure leads to a conditional association between the instrument and confounders. The initial proposal for non-linear Mendelian randomisation was to stratify on the residual exposure, defined as the residual from regression of the exposure on the instrumental variables.[13] The residual exposure is independent of the instrumental variables by construction, and the instrumental variables will be independent of confounders within strata of the residual exposure under a 'constant genetic effect' assumption.[15] However, if the effect of the genetic instrumental variables on the exposure varies between individuals, then this residual stratification method can lead to severe bias in stratum-specific estimates.[16 17]

An alternative method, known as the doubly-ranked method, allows the instrumental variable effects on the exposure to vary between individuals provided that a weaker 'rank-preserving assumption' holds.[18] The rank-preserving assumption states that the ranking of individuals according to their exposure levels would be the same

at all levels of the instrumental variable. That is, if the genotype of all individuals were set to any fixed value, the order of individuals according to their exposure levels would be the same. The doubly-ranked method is implemented by first ranking individuals according to their level of the instrument to form prestrata with similar values of the instrument, and then ranking individuals within each prestratum according to their level of the exposure, so that the lowest stratum contains the individual with the lowest exposure value from prestratum 1, the individual with the lowest exposure value from prestratum 2, the individual with the lowest exposure value from prestratum 3 and so on. This method has been shown to be more reliable than the residual method in simulation studies.[17 18]

In a previous paper, we presented evidence for the effect of BMI on mortality from non-linear Mendelian randomisation using the residual stratification method in the Norwegian Trøndelag Health (HUNT) and UK Biobank studies. A J-shaped association between genetically predicted BMI and mortality was observed in each study population overall, although a monotonic increasing association (ie, an always-increasing association) was seen in never-smokers.[19] Here, we present a reanalysis of the same data sets using the doubly-ranked stratification method. We first investigate whether the constant genetic effect assumption holds for this example, and explore the validity of the instrumental variable assumptions in strata of the population. We then present non-linear Mendelian randomisation findings from the doubly-ranked stratification method.

## METHODS

We present abbreviated descriptions of the data sets included in the analysis; detailed descriptions are in the original paper.[19]

### The HUNT study

We used data from the second wave (1995–1997) of the HUNT study on 65 229 individuals living in the northern part of Trøndelag and over the age of 20 years.[20] Participants were followed up until 15 June 2023 or their date of death. The original paper only considered mortality outcomes up to April 2015. By extending the follow-up period, we increase the number of mortality events considered in the analysis from 12 015 to 18 836. We excluded participants without data on BMI or genetic variants for BMI, leaving 56 150 individuals for analysis.

### The UK Biobank study

The UK Biobank cohort comprises around 500 000 participants (94% of self-reported European ancestry) aged 40–69 years at baseline. They were recruited between 2006 and 2010 in 22 assessment centres throughout the UK, and followed up until 28 February 2021 or their date of death.[21] Again, the original paper only considered mortality outcomes up to February 2016. By extending the follow-up period, we increase the

number of mortality events considered in the analysis from 10 344 to 25 021. We performed detailed quality control procedures on UK Biobank participants and on genetic variants as described previously.[22] In total, 366 385 unrelated European ancestry participants were included in analyses.

## SNPs and genetic score used as instrumental variables

77 single nucleotide polymorphisms (SNPs) were selected as candidate instrumental variables for BMI based on European sex-combined analyses in a genome-wide association study of the Genetic Investigation of ANthropometric Traits (GIANT) consortium.[23] Two of these variants (rs12016871 and rs2033732) were not available in the HUNT study, and a further two variants (rs13021737 and rs16951275) were excluded from the analyses due to association with smoking status in the HUNT study. An externally weighted score was calculated for each individual by multiplying the number of BMI-increasing alleles the individual carries by the variant's association with BMI from the GIANT study (online supplemental table 1), and summing across the remaining 73 variants. Overall, the genetic score explained 2.0% and 1.6% of the variance in BMI in the HUNT and UK Biobank studies, respectively, corresponding to F-statistics of 1121 and 5964. As a supplementary analysis, we repeated analyses for all-cause mortality using an extended genetic score comprising 633 variants taken from a more recent genome-wide association study of the GIANT consortium, which included UK Biobank.[24] The variant list was taken from Verkouter *et al*,[25] who pruned the initial set of reported variants to exclude correlated variants.

We assessed the instrumental variable assumptions within strata by estimating associations between the genetic score and various traits that are competing risk factors: smoking status (ever vs never), alcohol status (current vs other), education level (postsecondary or higher vs other), occupation (currently employed vs other), age at recruitment, and sex.

## Study design

We performed non-linear Mendelian randomisation analyses to estimate the shape of the association between genetically predicted BMI and the outcome. Our primary analysis considered all-cause mortality as the outcome. We also conducted subgroup analyses considering men and women separately. In addition, we studied associations with cause-specific mortality events (cardiovascular, cancer and non-cardiovascular non-cancer) in UK Biobank. These were defined based on International Classification of Diseases, Tenth Revision (ICD-10) codes (see online supplemental table 2 for details). We did not perform analyses stratified by smoking status (as presented in the original paper), as smoking status is influenced by BMI,[26] and hence this stratification may introduce collider bias.

## Statistical analyses

To estimate the non-linear relationship between genetically predicted BMI and mortality risk, a fractional polynomial method was applied.[12] In brief, first we divided the sample into 100 strata using the doubly-ranked method.[18] For comparison, we also present results from stratification using the residual method for the primary outcome of all-cause mortality. We then calculated the linear Mendelian randomisation estimate, referred to as a localised average causal effect (LACE), in each stratum of the population as a ratio of coefficients: the association of the genetic score with the outcome divided by the association of the genetic score with the exposure. Associations with the exposure (BMI) were obtained from linear regression; associations with the outcome (mortality) were obtained from Cox proportional hazards regression, using age as the timescale. While Cox proportional hazards regression is a semiparametric method, the estimated non-parametric baseline hazard function is a function of the timescale, not the exposure value. All associations were adjusted for age, age-squared, sex and centre (for UK Biobank).

We performed meta-regression of the LACE estimates against the mean of the exposure in each stratum in a flexible semiparametric framework using the derivative of fractional polynomial models of degrees 1 and 2. Under the Mendelian randomisation assumptions, the slope of the curve represents the average causal effect of the exposure on the outcome for the stratum with that exposure level. A positive causal effect is evident at a particular exposure level when the lower and upper confidence limits for the curve both have a positive slope; and similarly an inverse causal effect is evident if both confidence limits have a negative slope. The reference point in analyses is set to 25 kg/m$^2$; however, this reference point is arbitrarily chosen, and the key indicator of an increasing or decreasing effect is not whether CIs include or exclude the reference point, but rather whether the slope of the curve is positive or negative at a given exposure value.

We report two tests for non-linearity: a linearity test, which assesses whether a non-linear model fits the LACE estimates better than a linear model, and a trend test, which tests for a linear trend among the LACE estimates. Rejection of the linearity test indicates that a linear model is a suboptimal fit to the data; failure to reject means any improvement in fit for the best-fitting non-linear model over the linear model is no more than would be expected due to chance alone.

As LACE estimates from the doubly-ranked method can be sensitive to specification of the analytic sample, we repeated analyses 100 times for each data set omitting a small number of individuals in each iteration (12 individuals were removed at random in each iteration), and then combined estimates across iterations using Rubin's rules.

All statistical analyses were performed in R (V.4.3.1), and non-linear Mendelian randomisation analyses were performed using the SUMnlmr package.[27]

Table 1 Baseline characteristics of study participants in HUNT and UK Biobank studies

| | HUNT | UK Biobank |
|---|---|---|
| Number of participants (n) | 56 150 | 366 385 |
| Men (%) | 47.1 | 45.9 |
| Age at baseline in years (SD) | 49.6 (16.6) | 56.7 (8.0) |
| Number of deaths (n) | 18 836 | 25 021 |
| Cardiovascular mortality events (n) | – | 5212 |
| Cancer mortality events (n) | – | 12 880 |
| Other (non-cardiovascular non-cancer) events (n) | – | 6692 |
| BMI in kg/m$^2$ (SD) | 26.3 (4.1) | 27.4 (4.8) |
| Median follow-up years | 26.5 | 12.0 |
| Ever-smokers (%) | 55.9 | 46.1 |

Data are given as number of subjects (n), percentage (%), median or mean (SD). For 237 deaths in UK Biobank, sufficient data were not available to classify the event as cardiovascular, cancer or non-cardiovascular non-cancer. BMI, body mass index.

## Patient and public involvement

No patients were involved in setting the research question or the outcome measures, nor were they involved in the design or implementation of the study. No patients were asked to advise on interpretation or writing up of results. There are no specific plans to disseminate the results of the research to study participants or the relevant patient community.

## RESULTS

### Study populations

Baseline characteristics of the participants are provided in table 1. Genetic associations with BMI in strata of the population defined by the doubly-ranked method are displayed in online supplemental figure 1, and the BMI distribution in strata of the population is given in online supplemental table 3. Per 1 SD increase in the genetic score, BMI was 0.358 kg/m$^2$ greater in the lowest percentile of participants in the HUNT study, and 1.126 kg/m$^2$ greater in the highest percentile. Equivalent values for UK Biobank were 0.232 kg/m$^2$ in the lowest percentile and 1.645 kg/m$^2$ in the highest percentile. It is clear that the genetic associations with BMI differ strongly at different levels of BMI, and so the assumption required for the residual stratification method is violated. We note that similar estimates of the genetic associations with the exposure from the residual stratification method are unreliable, as these estimates from the residual method are similar even if the genetic effect on the exposure varies in the population.[18]

### Assessment of instrument validity

Associations with traits that are competing risk factors in strata of UK Biobank are displayed in figure 1. As previously observed in the literature for other exposures,[28] there are clear patterns in genetic associations with age and sex across strata. While there is no overall association of the genetic score with age or sex, there are associations in several of the strata.

As it is logically impossible that autosomal genetic variants can affect either age or sex, and as age and sex are typically measured without error, these associations must represent the effect of selection into the UK Biobank sample. While concerning, as we adjust for age and sex in our Mendelian randomisation analyses, these associations will not lead to strong bias in our causal estimates (see online supplemental material for a simulation study investigation into bias mitigation by adjustment for confounders that are predictors of selection). For other traits, both the magnitude of associations and patterns in associations are less strong, particularly for education level and occupation, which are important measures of social class. For alcohol status and smoking status, it is impossible to distinguish between genetic associations that reflect pleiotropic or selection effects and those that reflect a causal effect of the exposure (sometimes called 'vertical pleiotropy'); the latter do not violate the instrumental variable assumptions. Hence, both the associations and any pattern of associations may reflect violations of the instrumental variable assumptions or selection bias, but they could also reflect downstream effects of BMI which vary across strata.

Corresponding associations in the HUNT study are shown in online supplemental figure 2. While there are some patterns in the stratum associations with age and sex, these are less pronounced than for UK Biobank, potentially due to differential selection being less strong in the HUNT study.

### Comparison of results from residual and doubly-ranked methods

The shape of the association between genetically predicted BMI and all-cause mortality is displayed in figure 2, and stratum-specific LACE estimates are provided in online supplemental figure 3. We note that figure 2 and online supplemental figure 3 differ in their presentation of these data; figure 2 plots the estimated association between genetically predicted BMI values and mortality risk with respect to the reference exposure value, whereas online supplemental figure 3 plots the estimated effect of the exposure on mortality risk in each stratum. A smoothed version of the log transformed estimates in online supplemental figure 3 would represent the derivative of the curve shown in figure 2.

When stratifying the population using the residual method, there is a distinct J-shaped relationship between BMI and mortality, particularly in the UK Biobank study, with minimum risk at a BMI level of around 25 kg/m$^2$. However, as discussed above, results from the residual

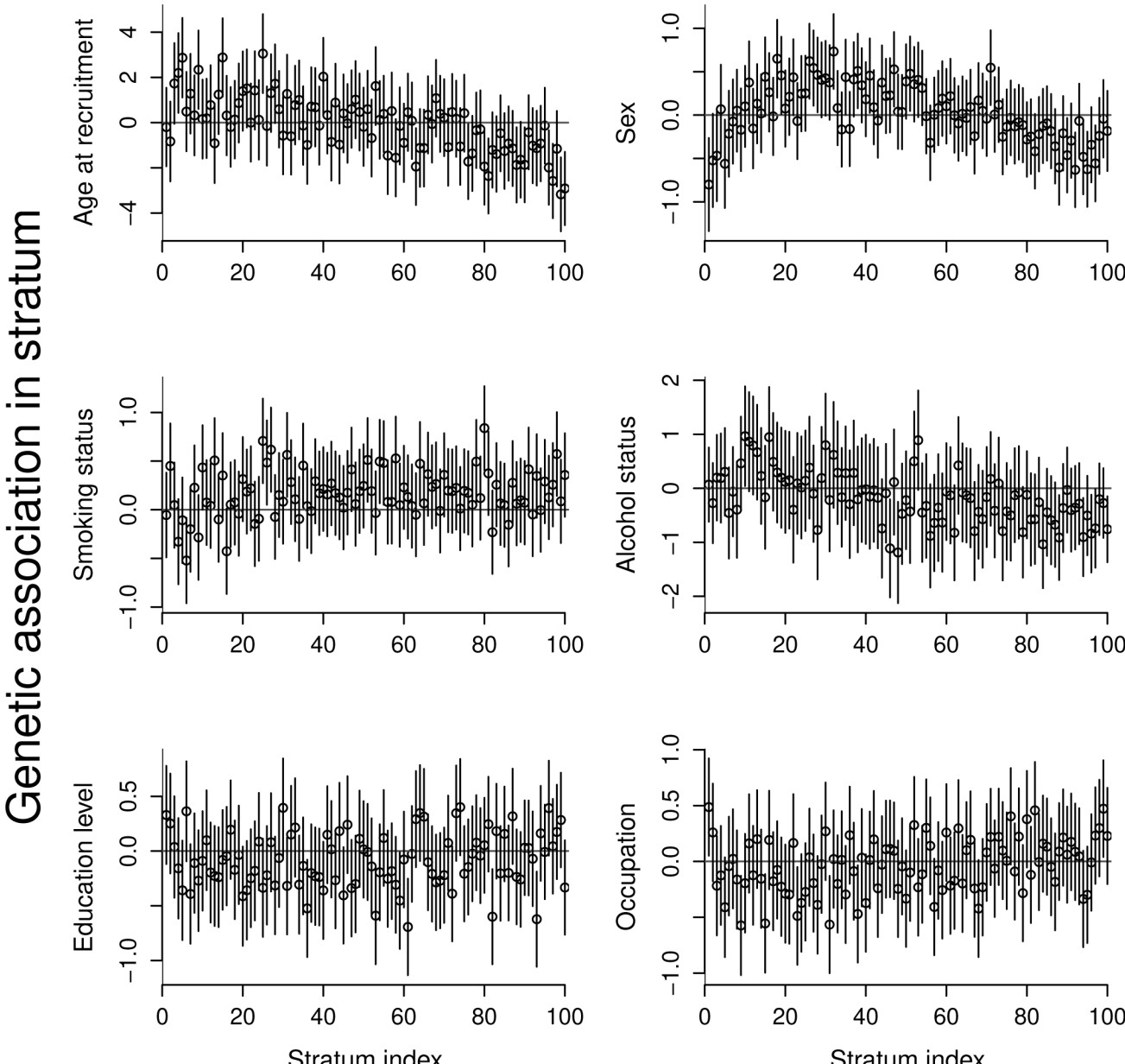

**Figure 1** Genetic associations with potential competing risk factors in strata of the UK Biobank population.

stratification method are unreliable here. In contrast, when stratifying using the doubly-ranked method, the relationship between BMI and mortality is far more uncertain at the lower end of the BMI distribution. In the HUNT study, the association is mildly J-shaped, with a null slope up to a BMI of around 28 kg/m², and a positive slope above this level. For UK Biobank, there is some indication of an upturn in mortality risk at very low levels of BMI (the slope of the curve is negative below 20 kg/m²), but otherwise the relationship is flat (and compatible with a null effect) up to a BMI of around 25 kg/m², and has an increasing positive slope above this level. The linearity and trend tests for the residual method indicate evidence supporting non-linearity for UK Biobank ($p_{linearity}=5\times10^{-4}$, $p_{trend}=2\times10^{-5}$). In contrast, these tests for the more reliable doubly-ranked method do not provide compelling evidence supporting non-linearity (figure 2).

Similar results were observed when inverse-normal rank transforming the exposure (online supplemental figure 4). When dividing the population into 50 strata, results were similar in both studies. When dividing into 200 strata, results were similar for UK Biobank, but differed for the HUNT study, as the best-fitting curve was monotone with no J-shaped upturn at the lower end of the BMI distribution (online supplemental figure 5). When using the extended genetic score (633 variants) in UK Biobank, a pattern of results similar to the main analysis was observed, but with narrower CIs as the extended score explains more variance in BMI (online supplemental figure 6). There was still some indication of a J-shaped curve, although minimum mortality risk was at a BMI level of 20–21 kg/m², and evidence for an upturn in risk at low BMI levels was weaker.

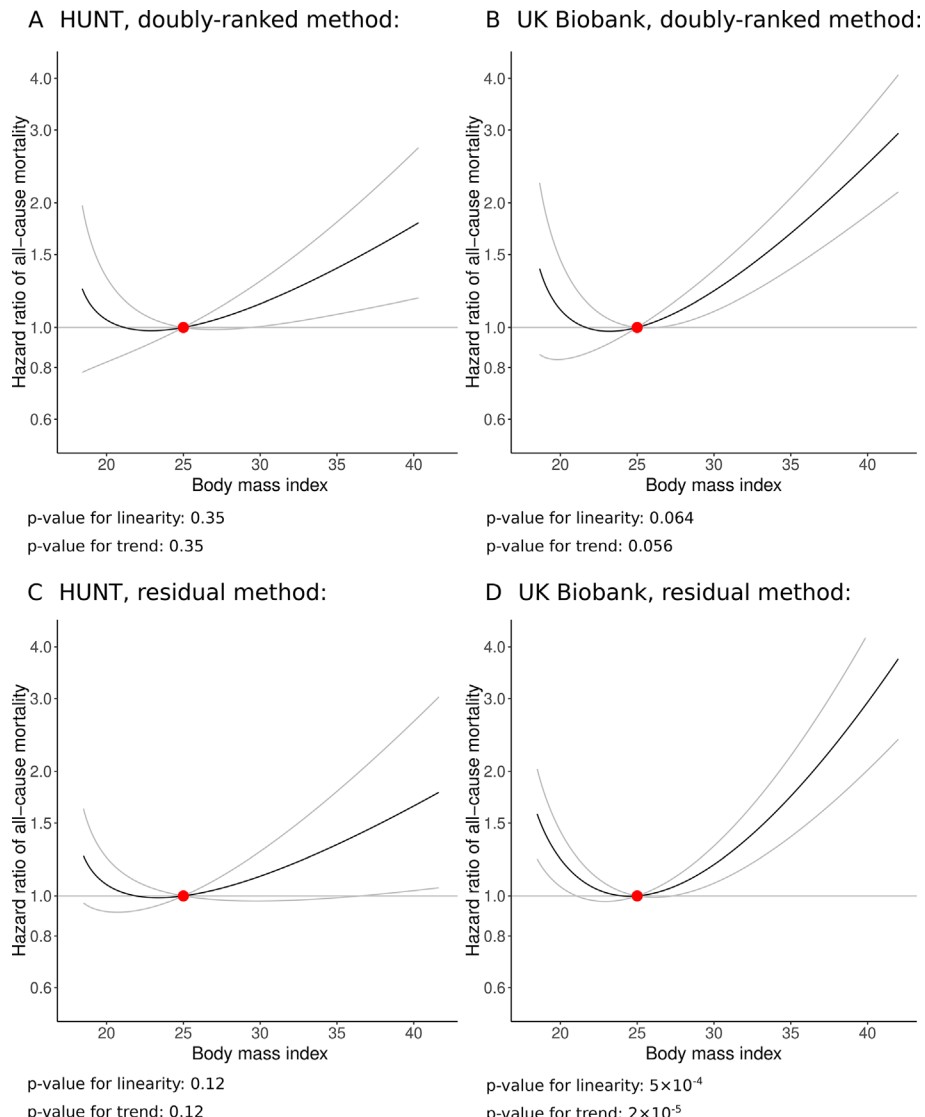

**Figure 2** Non-linear Mendelian randomisation—dose–response curve between body mass index and all-cause mortality. Using the doubly-ranked method: (A) HUNT, (B) UK Biobank; using the residual method (unreliable when the genetic effect on the exposure varies); (C) HUNT and (D) UK Biobank. Gradient at each point of the curve is the localised average causal effect. Grey lines represent 95% CIs. The reference value for BMI was taken as 25 kg/m$^2$.

## Sex-stratified analyses

Sex-stratified analyses from the doubly-ranked method are presented in figure 3, and stratum-specific LACE estimates are provided in online supplemental figure 7. In each case, there is some indication of a J-shaped relationship, but an upturn in mortality risk is only evident below 20 kg/m$^2$ (below 22 kg/m$^2$ for men in the HUNT study). Otherwise, estimates are generally compatible with the null up to a BMI of 25 kg/m$^2$, and positive above this level. Evidence for a harmful effect of increases in BMI for both men and women is weaker in the HUNT study, although a harmful effect above 30 kg/m$^2$ is evident for men. The slope above 25 kg/m$^2$ is steeper for women than men in UK Biobank. The linearity and trend tests do not provide compelling evidence supporting non-linearity in any analysis (figure 3).

## Cause-specific mortality

Analyses investigating cause-specific mortality from the doubly-ranked method in UK Biobank are presented in figure 4, and stratum-specific LACE estimates are provided in online supplemental figure 8. Again, there is some indication of a J-shaped relationship for each outcome. This is most pronounced for cardiovascular mortality, and less evident for cancer and non-cardiovascular non-cancer mortality. An upturn in risk at low BMI levels was only evident for cardiovascular mortality, and only below 20 kg/m$^2$. Otherwise, the relationship was compatible with the null up to around 24 kg/m$^2$ for cardiovascular mortality, 26 kg/m$^2$ for non-cardiovascular non-cancer mortality and 30 kg/m$^2$ for cancer mortality, and positive above this level. The slope above 25 kg/m$^2$ is steepest for cardiovascular mortality, then non-cardiovascular non-cancer, and finally for cancer mortality. The linearity and

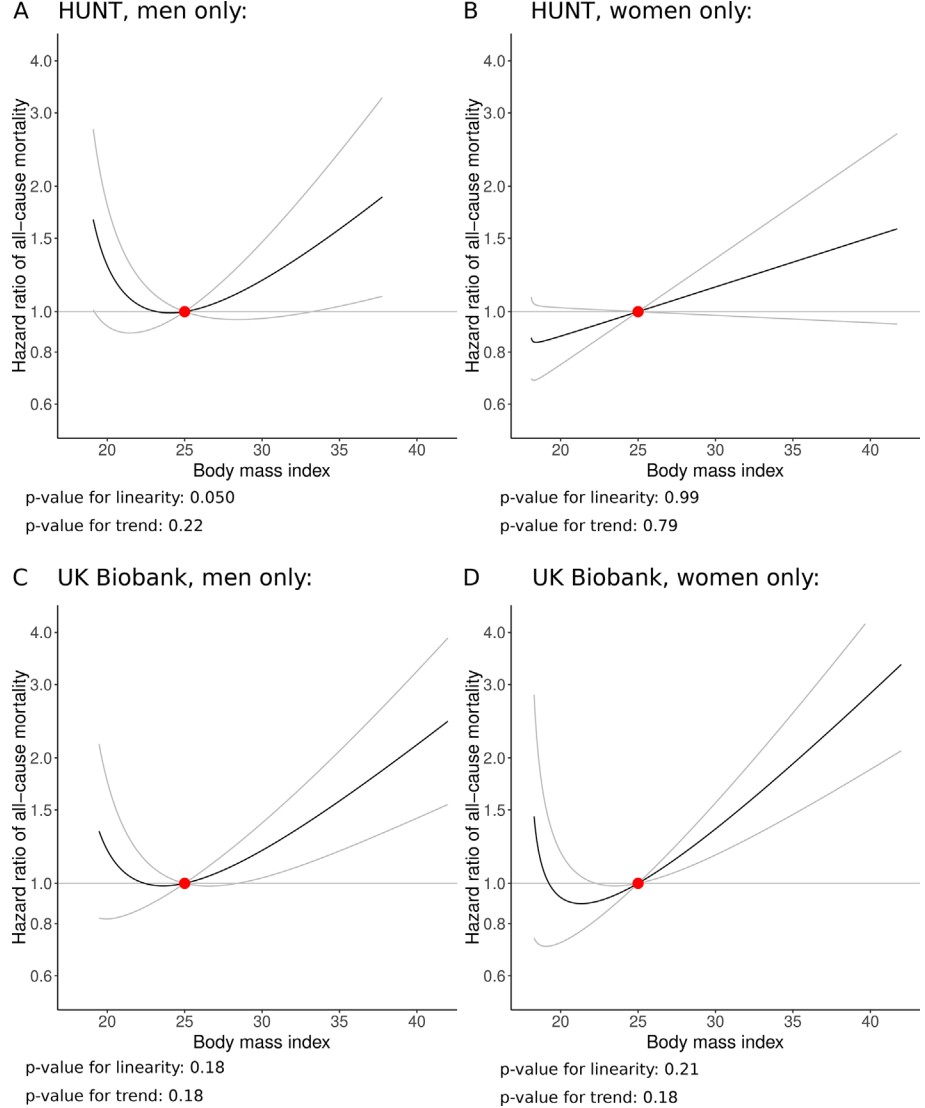

**Figure 3** Non-linear Mendelian randomisation—dose–response curve between body mass index and all-cause mortality in men and women, respectively, using doubly-ranked method. (A) HUNT, men only; (B) HUNT, women only; (C) UK Biobank, men only (D) UK Biobank, women only. Gradient at each point of the curve is the localised average causal effect. Grey lines represent 95% CIs. The reference value for BMI was taken as 25 kg/m². BMI, Body Mass Index.

trend tests do not provide any evidence supporting nonlinearity for any outcome, indicating that a non-linear model does not fit the data significantly better than a linear model for any outcome (figure 4).

## DISCUSSION

In this manuscript, we have investigated the shape of associations between genetically predicted BMI and mortality outcomes to provide an indication of the causal effect of BMI on mortality risk at different levels of BMI using non-linear Mendelian randomisation. While the curves display some signs of a J-shaped relationship, evidence for a harmful average effect of decreased BMI on mortality at low BMI levels was limited. Evidence supporting non-linearity from statistical hypothesis tests for the doubly-ranked method was unconvincing for all mortality outcomes. In UK Biobank, the slope representing the

estimated effect of BMI on mortality risk above 25 kg/m² was sharper for women than for men, and sharpest for cardiovascular mortality compared with cancer mortality and non-cardiovascular non-cancer mortality. There was evidence for a harmful effect of high BMI on all mortality subtypes.

Our findings challenge 'obesity paradox' results from observational epidemiology and previous Mendelian randomisation analyses using the residual method, which suggest that increases in BMI might reduce risk of mortality for a substantial proportion of the general population. While the minimum of the estimated curve for all-cause mortality was at a BMI of around 23 kg/m² in UK Biobank, the curve was near flat from 22 to 24 kg/m², and was positive from 25 kg/m² upwards. Evidence for a harmful effect of decreased BMI at low levels of BMI (as indicated by negative slopes for both the upper and

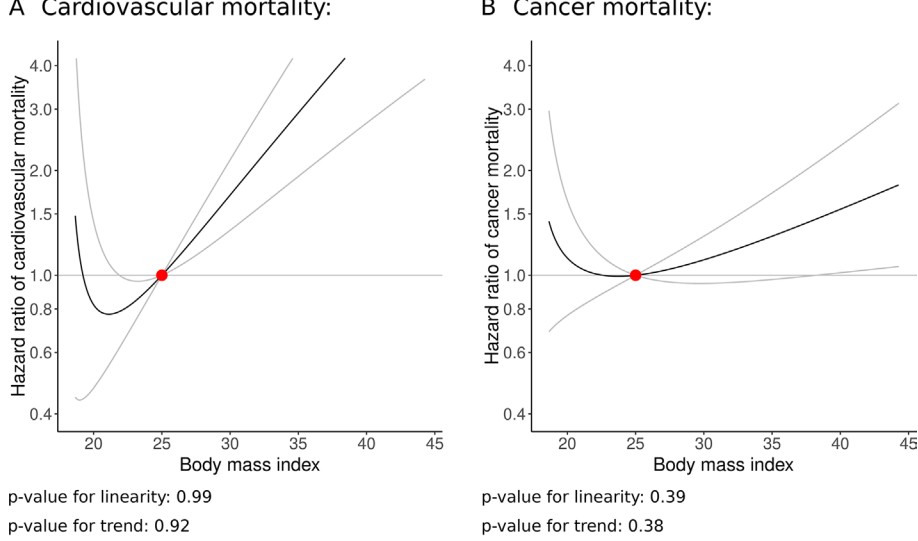

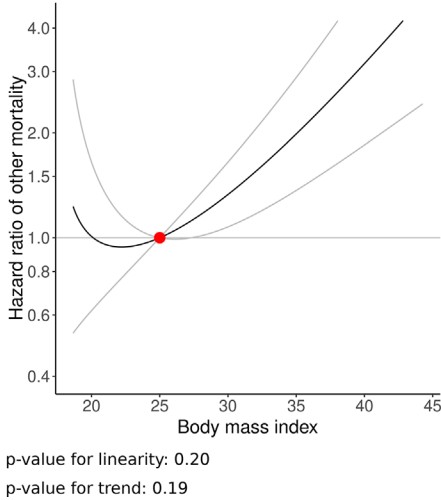

**Figure 4** Non-linear Mendelian randomisation in UK Biobank—dose–response curve between body mass index and cause-specific mortality using doubly-ranked method. (A) Cardiovascular disease mortality; (B) cancer mortality and (C) mortality due to other causes (non-cardiovascular, non-cancer). Gradient at each point of the curve is the localised average causal effect. Grey lines represent 95% CIs. The reference value for BMI was taken as 25 kg/m².

lower confidence limits of the BMI—mortality curve) was only present in some analyses, and only below 20 kg/m² when present. Among cause-specific mortality outcomes, this was only evident for cardiovascular mortality. In the HUNT study, there was no strong evidence for an effect of BMI on all-cause mortality at low BMI levels in overall analyses, and evidence for a harmful effect of increased BMI above around 28 kg/m². A potential reason for the discrepancy is that UK Biobank only contains participants from the age of 40 upwards, whereas the HUNT study includes younger individuals. Hence, underweight strata in UK Biobank are more likely to comprise primarily of older, frailer individuals, whereas underweight strata in the HUNT study include more healthy, slim individuals.

Our investigation provides further empirical evidence that the assumptions required for the residual stratification method for non-linear Mendelian randomisation are

likely to be violated in practice, and that results from the residual method can be substantially different to those from the doubly-ranked method.[17] The ratio between the genetic association with BMI in the highest versus lowest percentile group was 3.1 in the HUNT study and 7.1 in UK Biobank, whereas the assumptions required by the residual method imply that these associations should be constant across the distribution of the exposure. Particularly in UK Biobank, the residual method provided strong evidence for non-linearity in the effect of BMI on mortality risk, giving a J-shaped relationship with the minimum risk level at around 25 kg/m². This is similar to the confounded association observed in traditional observational studies.[2 3] In contrast, the doubly-ranked method provided little evidence for non-linearity; formal statistical tests for non-linearity did not reject the null hypothesis of linearity in any analysis. While there was some indication

of a J-shaped curve in the main analyses, estimates at the lower end of the BMI distribution were imprecise. One reason for this is that genetic associations with BMI were weaker at low levels of BMI in the doubly-ranked method, and so stratum-specific Mendelian randomisation estimates are less precise in these strata. In some supplementary analyses using different numbers of strata, evidence for a J-shaped relationship in the HUNT data set was absent and the J-shape was less pronounced when using an extended genetic score in UK Biobank.

Previous investigations into the non-linear shape of the causal relationship between BMI and mortality have used a variety of approaches. Wade *et al* used the residual stratification method for non-linear Mendelian randomisation in UK Biobank,[29] which showed a J-shaped relationship similar to the one we observe here. Jenkins *et al* investigated associations between a genetic score for BMI and mortality in UK Biobank,[5] considering analyses in the full data set, and restricted to those with a pre-existing morbidity condition. While they did not observe evidence for non-linearity, the genetic score only explains a small proportion of variance in BMI, and so substantial non-linearity may not be expected on this scale. The difference between average BMI levels at the lowest versus highest decile of the genetic score is less than 2 kg/m$^2$. Sulc *et al* investigated the effect of BMI on life expectancy (estimated as the mean of parental lifespan) in a Mendelian randomisation framework using a polynomial approach similar to the residual method,[30] except that instead of stratifying on the residual exposure, this approach constructs a polynomial function of the residual exposure. They showed a linear effect of increased BMI on reduced life expectancy, with no indication of non-linearity. However, this approach also has been shown in simulations to be vulnerable to variation in genetic effects on the exposure.[18] Carslake *et al* performed an instrumental variable analysis to investigate the impact of BMI on mortality in the HUNT study,[31] but using offspring BMI as an instrument for the BMI of study participants. This analysis did not provide evidence for a harmful effect of having low BMI. However, offspring BMI is unlikely to satisfy the instrumental variable assumptions for the exposure of participant BMI. A similar analysis was performed in the 1958 British birth cohort, and a similar conclusion was reached.[32] Blond *et al* also performed an instrumental variable analysis using offspring BMI as an instrument for the BMI of study participants in the Copenhagen School Health Records Register,[33] showing evidence of positive effects of increases in BMI on mortality at both high and low BMI levels. Finally, Kjøllesdal *et al* performed an instrumental variable analysis using early adulthood BMI as an instrument for the midlife BMI of study participants in various Norwegian health surveys,[34] showing mild J-shaped associations in the relationship between BMI and mortality, but far less pronounced than from observational analyses of the same dataset.

Overall, there is general consensus from different approaches that the adverse relationship of low BMI with mortality is exaggerated in observational analyses. However, variation between these results indicates the importance of modelling assumptions, both for the identification of causal effects, and for the modelling of non-linear relationships. Indeed, in our investigations, if we had used fractional polynomials of degree 1 rather than degree 2, then we would have observed monotonic increasing relationships of BMI with all outcomes. This is because the simpler degree 1 models are dominated by behaviour at the upper end of the BMI distribution, where instruments are strongest and most mortality outcomes occur. The simpler degree 1 polynomials smooth over imprecise negative estimates at the lower end of the BMI distribution, and so do not allow uncertainty in the shape of the distribution (particularly at its lower end) to be accurately reflected.

Our analysis has several strengths, but also limitations. The Mendelian randomisation design minimises bias from confounding and reverse causation. The large sample sizes of the HUNT and UK Biobank studies enable powerful analyses, even in strata of the population with low BMI levels. These sample sizes also enable a fine stratification of individuals, meaning that outcome associations can be assessed in strata only consisting of low BMI ($<20$ kg/m$^2$) individuals.

However, there are important limitations, which should lead to caution in the interpretation of our findings. As with all Mendelian randomisation analyses, findings are dependent on the validity of the genetic variants as instrumental variables. While we were able to show null associations of the genetic score with some traits, it is never possible to prove the validity of the instrumental variable assumptions beyond all doubt. To reduce the scope for population stratification, our analyses were limited to European ancestry participants. Our findings may therefore not be applicable to other populations. Furthermore, recruitment into UK Biobank is dependent on age, sex and other factors, which leads to bias in Mendelian randomisation estimates. While the effect of moderate selection on Mendelian randomisation estimates is often slight,[35] the extent of selection bias may differ between strata, leading to differential bias across stratum-specific estimates.[28] It is likely that age and sex are the strongest predictors of study participants, and differential genetic associations with these traits were observed in UK Biobank. For other traits, patterns in genetic associations across strata were less evident, and may reflect downstream effects of BMI rather than instrument invalidity due to pleiotropy or selection bias. As the HUNT study focused on a specific geographic area of Norway, it achieved much higher recruitment rates (around 70%, compared with 5% in UK Biobank), and so should be less affected by selection bias. This was corroborated by our analyses, as genetic associations in strata were weaker in the HUNT study. While genetic associations are protected from reverse causation, strata membership is determined by BMI levels, and hence could be subject to reverse causation. In particular, low BMI strata could

contain a large proportion of individuals whose BMI levels are low because of comorbidity. However, genetic associations should still provide a reliable guide as to the extent and direction of any causal effect of BMI in these strata. Finally, all estimates represent averages across strata of the population; individual effects of raising or lowering BMI may vary between individuals. Differences in estimates for different strata may reflect the changing composition of the strata, as the characteristics of those with BMI less than 20 kg/m$^2$ are likely to be different to those with higher BMI. A non-linear curve may reflect a combination of different effects in different subgroups of the population, rather than that the relationship between BMI and mortality risk is non-linear for any individual in the population.[15]

## CONCLUSIONS

In conclusion, non-linear Mendelian randomisation analyses using the doubly-ranked stratification method provide strong evidence for harmful effects of increased BMI on mortality above 25 kg/m$^2$. Evidence for a harmful effect of low BMI was only present in some analyses, and where present, only below 20 kg/m$^2$.

**Author affiliations**
[1]MRC Biostatistics Unit, University of Cambridge, Cambridge, UK
[2]Cardiovascular Epidemiology Unit, Department of Public Health and Primary Care, University of Cambridge, Cambridge, UK
[3]Victor Phillip Dahdaleh Heart and Lung Research Institute, University of Cambridge, Cambridge, UK
[4]Department of Clinical and Molecular Medicine (IKOM), Norges teknisk-naturvitenskapelige universitet, Trondheim, Norway
[5]Department of Pathology, Clinic of Laboratory Medicine, St. Olavs Hospital, Trondheim University Hospital, Trondheim, Norway
[6]Center for Oral Health Services and Research Mid-Norway (TkMidt), Trondheim, Norway
[7]Australian Centre for Precision Health, University of South Australia, Adelaide, South Australia, Australia
[8]Department of Public Health and Nursing, Norwegian University of Science and Technology, Trondheim, Norway

**Acknowledgements** The Trøndelag Health Study (HUNT) is a collaboration between the HUNT Research Centre (Faculty of Medicine and Health Sciences, NTNU, Norwegian University of Science and Technology), Trøndelag County Council, Central Norway Regional Health Authority and the Norwegian Institute of Public Health.

**Contributors** SB, Y-QS and X-MM conceptualised the project. SB and Y-QS performed formal analysis. SB, Y-QS, AZ, CB, AMM and X-MM contributed to the methodology and interpretation of findings. Y-QS and AMM helped with data curation. SB and X-MM supervised the project. SB wrote the initial manuscript draft. All authors (SB, Y-QS, AZ, CB, AMM and X-MM) contributed to the drafting and editing of the manuscript. All authors (SB, Y-QS, AZ, CB, AMM and X-MM) have approved of the final version of the manuscript. SB accepts full responsibility for the work and the conduct of the study, had access to the data, and controlled the decision to publish.

**Funding** This work was supported by the Wellcome Trust (grant number 225790/Z/22/Z), the UK Research and Innovation Medical Research Council (grant number MC_UU_00002/7), the British Heart Foundation (RG/18/13/33946) and the National Institute for Health and Care Research Cambridge Biomedical Research Centre (BRC-1215-20014; NIHR203312). YQS was supported by a researcher grant from The Liaison Committee for Education, Research and Innovation in Central Norway (project ID 2018/42794). AMM is funded by the National Institute for Health and Care Research Blood and Transplant Research Unit in Donor Health and Behaviour (NIHR203337). The views expressed are those of the authors and not necessarily those of the NIHR, NHSBT or the Department of Health and Social Care.

**Competing interests** None declared.

**Patient and public involvement** Patients and/or the public were not involved in the design, or conduct, or reporting or dissemination plans of this research.

**Patient consent for publication** Not applicable.

**Ethics approval** This study involves human participants. The HUNT study has ethical approval from The Regional Committee for Medical Research in Norway (REK 2015/78). Approval for individual projects is regulated in conjunction with The Norwegian Social Science Data Services (NSD). The UK Biobank study has ethical approval from the North West Multicentre Research Ethics Committee (16/NW/0274). This research has been conducted using the UK Biobank Resource under Application Number 7439. Participants gave informed consent to participate in the study before taking part.

**Provenance and peer review** Not commissioned; externally peer reviewed.

**Data availability statement** Data may be obtained from a third party and are not publicly available. Data from the HUNT and UK Biobank studies are available on application to any bona fide scientific researcher. Summary statistics for genetic associations with the exposure and outcome in strata for UK Biobank are available at http://doi.org/10.6084/m9.figshare.25041686.

**ORCID iDs**
Stephen Burgess http://orcid.org/0000-0001-5365-8760
Yi-Qian Sun http://orcid.org/0000-0002-9634-9236
Xiao-Mei Mai http://orcid.org/0000-0002-0426-7496

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
