## [Reviewer comments · BMJ Open]

ARTICLE DETAILS

TITLE (PROVISIONAL)	Body mass index and all-cause mortality in HUNT and UK Biobank studies: revised non-linear Mendelian randomization analyses
AUTHORS	Burgess, Stephen; Sun, Yi-Qian; Zhou, Ang; Buck, Christopher; Mason, Amy M; Mai, Xiao-Mei

VERSION 1 – REVIEW

REVIEWER	Heath, Alicia Imperial College London, School of Public Health
REVIEW RETURNED	10-Nov-2023

GENERAL COMMENTS	This study is a re-analysis of data previously reported for the shape of the causal relationship between BMI and mortality risk in a Mendelian randomization framework. The previous publication used the residual stratification method but the “constant genetic effect” assumption is violated because the genetic associations with BMI differ at different levels of BMI. In this new analysis, the doubly-ranked stratification method has been employed and, similar to the residual stratification method, indicated a J-shaped relationship, but it was less pronounced as estimates at the lower end of the BMI distribution were imprecise. The main finding is that above a BMI of 25 kg/m², risk of mortality increases with increasing BMI. BMI was also positively associated with cardiovascular mortality, cancer mortality, and non-cardiovascular non-cancer mortality; the association was sharpest for cardiovascular mortality. There was a lack of evidence of non-linearity. This is a well-conducted study with clearly presented results. A limitation is that there may be selection bias and this could vary across strata. However, the authors have done a good job of cautiously interpreting the results and discussing this potential bias, and in the supplement reported a simulation study to explore the issue of selection bias in a non-linear randomization framework. No method will ever be perfect, thus the authors are to be commended for seeking to use the current optimal method to answer the research question while acknowledging limitations. These new analyses provide us with the (current) best estimation of the shape of the relationship between BMI and mortality. Overall, the analyses have been conducted and reported suitably and the conclusions drawn are supported by the results. The findings of other studies have been appropriately summarized and interpreted in relation to the present findings. The manuscript is written very well and there are only a few minor points to address (detailed below).
--

	Abstract results: to retain the order mentioned in the methods, perhaps it would make more sense to report “for cancer mortality above 30 kg/m²” after cardiovascular mortality, before non-cardiovascular non-cancer mortality. Page 6: the outcomes cardiovascular mortality and cancer mortality could be defined more specifically (e.g. what ICD codes?) since there is often variation between studies in what is included within “cardiovascular” and “cancer”. Page 10: “there is some indication of an upturn in disease risk” – this is presumably meant to be mortality risk? Similarly for the sex-stratified analyses in “an upturn in disease risk is only evident below 20 kg/m²”. “An upturn in disease risk at low BMI levels was only evident for cardiovascular mortality” – saying disease risk might be confusing because here the risk is not the risk of incident cardiovascular disease, but rather risk of cardiovascular mortality. Omitting the word disease would avoid any confusion. Page 10: typo in “26 kg/m² for and non-cardiovascular” Page 12: “do not allow uncertainty in the shape of the distribution (particular at the lower end)” – meant to be particularly? Page 12: “findings are dependent on the validity of the genetic variants as instrumental variables” - if space in the word count allows, could you elaborate on this/put it in context of the present study – how valid are the genetic variants? Supplement page 2: typo (don’t need the a) in “This simulation study is a deliberately designed to be relatively simple”
--	--

REVIEWER	Kutalik, Zoltán University of Lausanne, Department of Computational Biology
REVIEW RETURNED	20-Nov-2023

GENERAL COMMENTS	Burgess et al present a reanalysis of the BMI-mortality relationship with somewhat updated data and applying a more robust non-linear MR method. The new method has weaker assumptions, hence its results are more reliable. However, the results are qualitatively not very different from the previous findings, simply their significance is lower and the conclusions are somewhat borderline: there might be some J-shaped trend, but there is no sufficient evidence for it. Overall, I do not find the new results representing a great advance and highlight lack of evidence rather than evidence for absence of a J-shaped relationship. So while the original paper needs an update (as it incorrectly concluded evidence for a J-shaped relationship), this current work would need more data to be more conclusive. Below I provide a couple of comments that could improve either the results or the presentation of the findings. Major comments: 1. Once sentence in the abstract does not accurately reflect the findings of the paper in my opinion:
---

	“stratification method still indicated a J-shaped relationship, but with less precision in estimates at the lower end of the BMI distribution” while in the Discussion it is admitted that “Evidence supporting non-linearity from the doubly-ranked method was unconvincing for all mortality outcomes.” I think the latter sentence is a more faithful description of the finding than the more positive (euphemistic) tone of the abstract (which is read by many more people than a nuanced discussion). 2. Supplementary Fig 3 (raw) and Fig 2 (smoothed) look strikingly different. Not much J-shaped trend can be observed by the original residual method and extremely modest, if any, trend is visible in Supp Fig 3 for the doubly-ranked method. Why do these trends look so markedly different, while the main figure is just a smoothed version of the Supplementary figure? In Supplementary Figure 3A&B, there are only 2-3 bins with BMI<20 and their estimates are all negative, I don’t know how it can provide any evidence for protective effect. If I look at the residual approach (C&D panels), the effects are even more negative, hence the original approach would provide even less evidence for a J-shaped relationship. Also, if I look at Fig 2, the curves by the two methods do not look qualitatively different, only the residual method has slightly steeper trends and less error. 3. When the outcome is a binary trait, and the exposure and outcome are linked through an already highly non-linear hazard function. Is there a special meaning of non-linearity? Or doesn’t it simply mean that using another hazard function we could potentially have flat HRs across strata? Applying their method to lifespan as a continuous outcome (restricting the analysis to the dead individuals) do the authors observe a non-linear causal effect? Do they see any non-monotonicity? 4. How sensitive are the results to using more (200) or less (50) bins? Also, since the previous study much larger number of instruments are available for BMI: it would be important to rerun the analysis with >900 instrument for BMI (https://pubmed.ncbi.nlm.nih.gov/30124842/). While there might be a small Winner’s curse, but since the exposure and outcome samples are the same, it should cancel out. These could at least be used for the HUNT study, where sample overlap is minimal. The results would be much more convincing if it stands when using more IVs. 5. I assume that the resulting smoothed LACE curves are unchanged if no age adjustment is done? (Since older study participants are genetically healthier, the relationship between BMI, age and mortality can be very intricate.) 6. If BMI is replaced with another measure of obesity, e.g. waist-to-hip ratio as exposure, do the conclusions remain the same? Minor comments: 1. “ranking of individuals according to their exposure levels would be the same at all levels of the instrumental variable” - The sentence is very vague and needs more clarification with a brief
--	---

	description of the two-stage ranking (first pre-strata by PRS, then strata via ranking by exposure). 2. “Sulc et al investigated the effect of BMI on life expectancy (estimated as the mean of parental lifespan) in a Mendelian randomization framework using a polynomial approach similar to the residual method²⁸, except that instead of stratifying on the residual exposure, this approach adjusts for the residual exposure.” – That method does not adjust for the residual exposure, but it adjusts for the exposure itself. Also, if the Sulc et al method had the same weakness as the original method by the authors (LACE), why does it find concordant results with the doubly-ranked new approach by the authors? 3. “using offspring BMI as an instrument for the BMI of study participant” – This is probably the worst instrument to imagine, likely violating most MR assumptions. Still seems to give similar result as a presumably more correct method. Can the authors comment on this? 4. Supplementary Fig 1A: why is there a sudden jump at BMI of 25 in the PRS-BMI association?
--	--

VERSION 1 – AUTHOR RESPONSE

Reviewer: 1

Comments to the Author:

A0. This study is a re-analysis of data previously reported for the shape of the causal relationship between BMI and mortality risk in a Mendelian randomization framework. The previous publication used the residual stratification method but the “constant genetic effect” assumption is violated because the genetic associations with BMI differ at different levels of BMI. In this new analysis, the doubly-ranked stratification method has been employed and, similar to the residual stratification method, indicated a J-shaped relationship, but it was less pronounced as estimates at the lower end of the BMI distribution were imprecise. The main finding is that above a BMI of 25 kg/m², risk of mortality increases with increasing BMI. BMI was also positively associated with cardiovascular mortality, cancer mortality, and non-cardiovascular non-cancer mortality; the association was sharpest for cardiovascular mortality. There was a lack of evidence of non-linearity.

This is a well-conducted study with clearly presented results. A limitation is that there may be selection bias and this could vary across strata. However, the authors have done a good job of cautiously interpreting the results and discussing this potential bias, and in the supplement reported a simulation study to explore the issue of selection bias in a non-linear randomization framework. No method will ever be perfect, thus the authors are to be commended for seeking to use the current optimal method to answer the research question while acknowledging limitations. These new analyses provide us with the (current) best estimation of the shape of the relationship between BMI and mortality.

> We appreciate the reviewer’s accurate summary of the contents of this paper, and its key messages. We appreciate her positive views on the submission, and in particular her comment that this represents the current best approach for estimating the shape of the relationship between BMI and mortality using a Mendelian randomization approach. <

Overall, the analyses have been conducted and reported suitably and the conclusions drawn are supported by the results. The findings of other studies have been appropriately summarized and interpreted in relation to the present findings. The manuscript is written very well and there are only a few minor points to address (detailed below).

A1. Abstract results: to retain the order mentioned in the methods, perhaps it would make more sense to report “for cancer mortality above 30 kg/m²” after cardiovascular mortality, before non-cardiovascular non-cancer mortality.

> We appreciate this comment, and have edited the text as suggested. <

A2. Page 6: the outcomes cardiovascular mortality and cancer mortality could be defined more specifically (e.g. what ICD codes?) since there is often variation between studies in what is included within “cardiovascular” and “cancer”.

> We have added a new table, Supplementary Table 2, which lists the precise ICD codes used to define cardiovascular and cancer mortality. We have added a brief sentence in the text: “These were defined based on International Classification of Diseases, Tenth Revision (ICD-10) codes (see Supplementary Table 2 for details).” <

A3. Page 10: “there is some indication of an upturn in disease risk” – this is presumably meant to be mortality risk? Similarly for the sex-stratified analyses in “an upturn in disease risk is only evident below 20 kg/m²”.

> We appreciate the reviewer’s close reading of the text. We have replaced “disease” with “mortality” at both points here. <

A4. “An upturn in disease risk at low BMI levels was only evident for cardiovascular mortality” – saying disease risk might be confusing because here the risk is not the risk of incident cardiovascular disease, but rather risk of cardiovascular mortality. Omitting the word disease would avoid any confusion.

> We have omitted the word “disease” as suggested. <

A5. Page 10: typo in “26 kg/m² for and non-cardiovascular”

> We have corrected this textual error to omit the word “and”. <

A6. Page 12: “do not allow uncertainty in the shape of the distribution (particular at the lower end)” – meant to be particularly?

> We have replaced with “particularly” as suggested. <

A7. Page 12: “findings are dependent on the validity of the genetic variants as instrumental variables” - if space in the word count allows, could you elaborate on this/put it in context of the present study – how valid are the genetic variants?

> We appreciate the suggestion. We have included a subsection in the Results section “Assessment of instrument validity”, which addresses this question in detail. We have added a sentence to the Discussion section: “While we were able to show null associations of the genetic score with some traits, it is never possible to prove the validity of the instrumental variable assumptions beyond all doubt.” <

A8. Supplement page 2: typo (don’t need the a) in “This simulation study is a deliberately designed to be relatively simple”

> We have corrected this textual error as suggested. <

Reviewer: 2

Comments to the Author:

B0. Burgess et al present a reanalysis of the BMI-mortality relationship with somewhat updated data and applying a more robust non-linear MR method. The new method has weaker assumptions, hence its results are more reliable. However, the results are qualitatively not very different from the previous findings, simply their significance is lower and the conclusions are somewhat borderline: there might be some J-shaped trend, but there is no sufficient evidence for it. Overall, I do not find the new results representing a great advance and highlight lack of evidence rather than evidence for absence of a J-shaped relationship. So while the original paper needs an update (as it incorrectly concluded evidence for a J-shaped relationship), this current work would need more data to be more conclusive. Below I provide a couple of comments that could improve either the results or the presentation of the findings.

> We appreciate the reviewer’s summary, although disagree with the reviewer’s assertion that there is no evidence for a J-shaped relationship. While the confidence intervals for the point estimate appear to include the null, the position of the null is dependent on the choice of reference exposure level,

which is an arbitrary choice. The key determinant signalling a negative slope at the lower end of the BMI distribution (and hence a J-shaped relationship) is whether the slope of the estimate and its confidence intervals are all negative. This is the case for UK Biobank in the overall analysis (doubly-ranked method) and the analyses split by sex, and for HUNT in the men only analysis (and there is a slight hint of an upturn in the women only analysis).

> This point is made in the manuscript: “A positive causal effect is evident at a particular exposure level when the lower and upper confidence limits for the curve both have a positive slope; and similarly an inverse causal effect is evident if both confidence limits have a negative slope. The reference point in analyses is set to 25 kg/m²; however, this reference point is arbitrarily chosen, and the key indicator of an increasing or decreasing effect is not whether confidence intervals include or exclude the reference point, but rather whether the slope of the curve is positive or negative at a given exposure value.” (page 7).

> However, as per the reviewer’s other comments, we accept that the evidence for a J-shaped relationship is not overwhelmingly strong (although we note in passing that the J-shape is stronger in Supplementary Figure 4 using inverse-normal rank transformed BMI values). We appreciate the reviewer’s comment that many readers will read the abstract only, and so we want this to be an accurate reflection of the findings of the paper, and the level of confidence we have in this finding. See the reply to point B1. <

Major comments:

B1. Once sentence in the abstract does not accurately reflect the findings of the paper in my opinion: “stratification method still indicated a J-shaped relationship, but with less precision in estimates at the lower end of the BMI distribution”

while in the Discussion it is admitted that “Evidence supporting non-linearity from the doubly-ranked method was unconvincing for all mortality outcomes.” I think the latter sentence is a more faithful description of the finding than the more positive (euphemistic) tone of the abstract (which is read by many more people than a nuanced discussion).

> We appreciate the reviewer’s close reading of the text. As per the reply to point B0, we believe there is some evidence for a J-shaped relationship. However, the statistical tests for non-linearity are non-significant, indicating that the best fitting non-linear model does not fit the data significantly better than the linear model. As per the reply to point B2, we believe that part of the reviewer’s scepticism towards a J-shaped relationship comes from a misinterpretation of Supplementary Figure 3 (and related supplementary figures).

> However, we acknowledge that evidence for a J-shaped relationship is not overwhelmingly strong. We have therefore amended the sentence in the abstract to read: “The re-analysis of these data using the more reliable doubly-ranked stratification method still indicated a potential J-shaped relationship, but with less certainty as there was less precision in estimates at the lower end of the BMI distribution.”. <

B2. Supplementary Fig 3 (raw) and Fig 2 (smoothed) look strikingly different. Not much J-shaped trend can be observed by the original residual method and extremely modest, if any, trend is visible in Supp Fig 3 for the doubly-ranked method. Why do these trends look so markedly different, while the main figure is just a smoothed version of the Supplementary figure? In Supplementary Figure 3A&B, there are only 2-3 bins with BMI<20 and their estimates are all negative, I don’t know how it can provide any evidence for protective effect. If I look at the residual approach (C&D panels), the effects are even more negative, hence the original approach would provide even less evidence for a J-shaped relationship. Also, if I look at Fig 2, the curves by the two methods do not look qualitatively different, only the residual method has slightly steeper trends and less error.

> The reviewer is mistaken in his interpretation of the difference between Supplementary Figure 3 and Figure 2. Figure 2 is not simply a smoothed version of Supplementary Figure 3. Figure 2 plots the relationship between the exposure and outcome – the y-axis is the hazard ratio of the outcome with respect to the reference exposure value. Supplementary Figure 3 plots the localized average causal effect (LACE) estimates of the exposure on the outcome – the y-axis is the LACE estimate, representing the estimated effect of the exposure on the outcome in that stratum. A smoothed curve

based on the estimates in Supplementary Figure 3 would be the derivative of the curve in Figure 2 (in less mathematical terms, the slope of the curve). A J-shaped relationship in Figure 2 would be represented by negative LACE estimates for low values of the exposure, and positive LACE estimates for greater values of the exposure. This is exactly what the reviewer sees in Supplementary Figure 3 (“2-3 bins with BMI<20 and their estimates are all negative”). Similarly, the observation that the reviewer makes for the residual method (“the effects are even more negative”) is exactly what would be expected if there were a stronger J-shaped relationship from the residual method, as we claim.

> We have clarified the interpretation of the LACE plots in Supplementary Figures 3, 5, and 6 [now Supplementary Figures 3, 7, and 8 in the revision] in the relevant figure captions, and in the main text: “We note that Figure 2 and Supplementary Figure 3 differ in their presentation of these data; Figure 2 plots the estimated association between genetically-predicted BMI values and mortality risk with respect to the reference exposure value, whereas Supplementary Figure 3 plots the estimated effect of the exposure on mortality risk in each stratum. A smoothed version of the plot in Supplementary Figure 3 would represent the derivative of the curve shown in Figure 2.”.

> We agree with the reviewer’s interpretation that the main differences between results from the two methods in Figure 2 are that the residual method has slightly steeper trends and less uncertainty, particularly at the lower end of the BMI distribution. <

B3. When the outcome is a binary trait, and the exposure and outcome are linked through an already highly non-linear hazard function. Is there a special meaning of non-linearity? Or doesn’t it simply mean that using another hazard function we could potentially have flat HRs across strata? Applying their method to lifespan as a continuous outcome (restricting the analysis to the dead individuals) do the authors observe a non-linear causal effect? Do they see any non-monotonicity?

> In Cox proportional hazards modelling, we fit a parametric component and a non-parametric component. The non-parametric component is the baseline hazard function. This is a function of time, which as the reviewer states, may be highly non-linear. This component is generally regarded as nuisance. The parametric component is a term including the model covariates, and is typically a linear predictor. We use coefficients from this linear predictor in our downstream analyses. As the non-parametric hazard function is a function of time, it is different from the non-linear function that we estimate here. The non-linear function that we estimate here is a function of the mean exposure value in the strata.

> We believe that restricting the analysis to dead individuals would not be a sensible idea, as it would induce selection bias that would render results as meaningless. As we use age as the time variable in our analysis, we already measure effects on lifespan in our current analysis.

> We have clarified that the non-parametric hazard function in Cox regression is a function of time: “While Cox proportional hazards regression is a semi-parametric method, the estimated non-parametric baseline hazard function is a function of the timescale, not the exposure value.”. <

B4. How sensitive are the results to using more (200) or less (50) bins? Also, since the previous study much larger number of instruments are available for BMI: it would be important to rerun the analysis with >900 instrument for BMI (<https://pubmed.ncbi.nlm.nih.gov/30124842/>). While there might be a small Winner’s curse, but since the exposure and outcome samples are the same, it should cancel out. These could at least be used for the HUNT study, where sample overlap is minimal. The results would be much more convincing if it stands when using more IVs.

> We have repeated overall analyses in both UK Biobank and HUNT with 200 and 50 strata (Supplementary Figure 5). For UK Biobank, results are indistinguishable from those with 100 strata. For HUNT, results are the same with 50 strata, but somewhat different with 200 strata – the smoothing method selects a monotone function rather than a J-shaped function. If we repeated analyses with 10 strata, we may expect the J-shape to be less prominent, as negative estimates were only observed in the lowest 3 strata (when dividing into 100). Hence, the estimate in the lowest stratum when dividing into 10 may not be negative, as it would average negative and null/positive estimates.

> While we appreciate that the extended genetic score would explain a greater proportion of variance in the exposure, including more variants in a Mendelian randomization analysis also leads to a greater

risk of pleiotropy. Additionally, we are reluctant to depart from our pre-specified analysis plan unless there is a strong reason to do this – the choice of instrument is a question of preference, not only that threatens the validity of the analysis. A further pragmatic reason is that this would require an additional data request to the HUNT study, which would delay the resubmission beyond the requested deadline. We note that 900+ is the number of variants identified using COJO – after pruning, there are around 650 uncorrelated variants identified as predictors of BMI by the paper cited above – we have performed analyses based on a previously reported list of 633 variants:

<https://www.ncbi.nlm.nih.gov/pmc/articles/PMC7660142/>.

> We have repeated analyses for UK Biobank using the extended score using these ~650 variants. However, we have retained the more limited score in main analyses as per the original analysis plan. Results are shown in Supplementary Figure 6. We see a similar pattern to results in main analyses, but with greater precision in estimates. The minimum of the BMI-mortality curve is shifted further to the left, at around 20-21 kg/m². There is still some indication of a J-shape for low BMI values, but evidence is not overwhelmingly convincing.

> We have added to the manuscript: “When dividing the population into 50 strata, results were similar in both studies. When dividing into 200 strata, results were similar for UK Biobank, but differed for the HUNT study, as the best-fitting curve was monotone with no J-shaped upturn at the lower end of the BMI distribution (Supplementary Figure 5). When using the extended genetic score (633 variants) in UK Biobank, pattern of results similar to the main analysis was observed, but with narrower confidence intervals as the extended score explains more variance in BMI (Supplementary Figure 6). There was still some indication of a J-shaped curve, although minimum mortality risk was at a BMI level of 20-21 kg/m², and evidence for an upturn in risk at low BMI levels was weaker.”. <

B5. I assume that the resulting smoothed LACE curves are unchanged if no age adjustment is done? (Since older study participants are genetically healthier, the relationship between BMI, age and mortality can be very intricate.)

> We are reluctant to perform analyses without adjustment for age for several reasons. Age is a strong determinant of mortality risk, and hence adjusting for age is important to correct for potential imbalances in the distribution of this variable within strata, and for consistency in estimates between strata. Additionally, previous investigations have shown that stratification can lead to genetic associations with age within strata (see <https://www.medrxiv.org/content/10.1101/2023.08.21.23293658v1>), which are important to adjust for. Hence, we do not see a justification for performing analyses without adjustment for age – such analyses would not reveal anything of interest, as any differences with the current results would likely be a reflection of bias. <

B6. If BMI is replaced with another measure of obesity, e.g. waist-to-hip ratio as exposure, do the conclusions remain the same?

> This is an interesting question, but is out of scope for the current manuscript. It would require a different set of genetic variants (together with necessary checks for validity), and would produce a complete set of different analysis results. Particularly as this goal of this submission is to update previously published BMI results using a more reliable method for non-linear Mendelian randomization, we would not want to complicate the picture further by adding new analyses. <

Minor comments:

B7. “ranking of individuals according to their exposure levels would be the same at all levels of the instrumental variable” – The sentence is very vague and needs more clarification with a brief description of the two-stage ranking (first pre-strata by PRS, then strata via ranking by exposure).

> We have edited the manuscript to provide a clearer explanation of both the rank-preserving assumption and the doubly-ranked method: “The rank-preserving assumption states that the ranking of individuals according to their exposure levels would be the same at all levels of the instrumental variable. That is, if the genotype of all individuals were set to any fixed value, the order of individuals according to their exposure levels would be the same. The doubly-ranked method is implemented by first ranking individuals according to their level of the instrument to form pre-strata with similar values of the instrument, and then ranking individuals within each pre-stratum according to their level of the

exposure, so that the lowest stratum contains the individual with the lowest exposure value from pre-stratum 1, the individual with the lowest exposure value from pre-stratum 2, the individual with the lowest exposure value from pre-stratum 3, and so on. This method has been shown to be more reliable than the residual method in simulation studies.”. <

B8. “Sulc et al investigated the effect of BMI on life expectancy (estimated as the mean of parental lifespan) in a Mendelian randomization framework using a polynomial approach similar to the residual method²⁸, except that instead of stratifying on the residual exposure, this approach adjusts for the residual exposure.” – That method does not adjust for the residual exposure, but it adjusts for the exposure itself. Also, if the Sulc et al method had the same weakness as the original method by the authors (LACE), why does it find concordant results with the doubly-ranked new approach by the authors?

> Thank you for this clarification. The PolyMR method adjusts for the exposure, and estimates a polynomial function of the residual exposure. We have corrected this point in the paper: “...except that instead of stratifying on the residual exposure, this approach constructs a polynomial function of the residual exposure”.

> As for why the PolyMR method gives different answers to the previous residual method, we could speculate many empirical reasons: different outcome (mortality versus lifespan), different instrument choice, different non-linear model, power considerations. However, we are not able to determine the reason with any degree of certainty. <

B9. “using offspring BMI as an instrument for the BMI of study participant” – This is probably the worst instrument to imagine, likely violating most MR assumptions. Still seems to give similar result as a presumably more correct method. Can the authors comment on this?

> Sometimes, even a stopped clock tells the right time. We would caution against judging the validity (or otherwise) of methods based on their performance in a single applied example.

> We have added to the text: “However, offspring BMI is unlikely to satisfy the instrumental variable assumptions for the exposure of participant BMI.”. <

B10. Supplementary Fig 1A: why is there a sudden jump at BMI of 25 in the PRS-BMI association?

> The prosaic explanation is that an unusually large number of individuals in the HUNT study have BMI values between 24.7 and 24.8. 543 individuals have BMI between 24.6 and 24.7, and 569 have BMI between 24.8 and 24.9. But 715 individuals have recorded BMI between 24.6 and 24.7. (For reference, the number of individuals in any single interval should follow a Poisson distribution, and a Poisson distribution with a mean of 600 would have a standard deviation of $\sqrt{600}$, which is just under 25. So these variations are beyond what would be expected due to chance alone: 544 is the 1st percentile of a Poisson distribution with mean 600, and 714 is the 99.9997th percentile.)

> The underlying explanation for this is not clear – perhaps there is some administrative reason why having a BMI value of 25 or below is desirable? Or perhaps individuals close to the “overweight” boundary were encouraged to remove an item of clothing and re-weigh themselves? (From the study protocol: “Height was measured with the participants wearing light clothes without shoes and given in whole centimeters. Weight was measured with the participants wearing light clothes without shoes.”) This could mean that some individuals with a true BMI just above 25 are recorded as having a BMI of 24.7. Or perhaps individuals regulate their diet to ensure that their BMI is close to 25, representing a potential weight loss goal. This would mean that the reported values of BMI are correct, even if this means the distribution is naturally coarsened.

> In any case, the impact of this anomaly in the BMI distribution of HUNT is likely negligible. The Gelman—Rubin statistics reported by the doubly-ranked method are all close to 1, indicating that the exposure is not too unevenly distributed to implement the doubly-ranked method. The statistical impact on results would be that LACE estimates around 24.7-24.8 could be slightly lower than they should be (if the genetic associations with BMI are overestimated), but this should have no substantive impact on the estimated BMI—mortality curve. <

VERSION 2 – REVIEW

REVIEWER	Heath, Alicia Imperial College London, School of Public Health
REVIEW RETURNED	20-Feb-2024

GENERAL COMMENTS	Thank you for responding to all comments and adding greater clarity to the manuscript. All comments have been suitably addressed and I have no further comments except for a few minor formatting errors to correct: Introduction first paragraph – missing units kg/m² for BMI Page 10 of revised manuscript: “The was still some indication of a J-shaped curve” – should be “There was...” In Figures 3 and 4 and Supplementary Figure 5, the x-axis of panel D has much fewer labels (only at BMI of 20, 30, 40), whereas the other panels in the figures use step points of 5 – it would be useful to add a few more labels/make it consistent e.g. 20, 25, 30, 35, 40, 45. I’m not sure if it is just something wrong with the way the PDF was rendered but Supplementary Figure 3 seems to have a superimposed label for panel D (with B and UK Biobank doubled under the text) Discussion: “Our analysis has several strengths, but also limitations” – might not need to say “but also limitations” here since the next paragraph says “However, there are important limitations”
---

REVIEWER	Kutalik, Zoltán University of Lausanne, Department of Computational Biology
REVIEW RETURNED	09-Feb-2024

GENERAL COMMENTS	I thank the authors for reassuringly addressing most of my points. I still stand by my opinion that the evidence shown here for a J-shaped relationship is largely insufficient and the main conclusion of the paper should rather be that a better method (and follow-up analysis) invalidates the original paper and more data is needed to reassuringly confirm a J-shaped relationship (even if there is one it is very mild). Having considered all pros and cons, I think it is still better to publish a new version of the retracted paper, even if it represents little evidence for a J-shaped relationship, but at least the analysis is done very carefully. I only have a few remaining outstanding comments I would like to insist on: In order to transparently present the pros and cons for the J-shaped relationship, I would like the authors briefly repeat all arguments that are not in support for a J-shaped relationship in one place in the Discussion. Namely,  1. When using 200 bins, the HUNT study shows no evidence for a J-shaped relationship. 2. The best fitting non-linear model is not significantly better than the linear causal model. (While a smoothed J-shaped curve looks nice, the truth is that a linear function would fit the data just as well.)
--

	3. Using the extended PRS in the UK Biobank leads to a perfectly linear trend for BMI >21, which represents >95% of the UKB population. Thus evidence for J-shape is even weaker than in the old PRS. This is a substantial shift compared to the minimum value at 25 in the original analysis (~35th percentile vs 5th). 4. Even with 100 bins, in HUNT, the slope in null up to a BMI of 28. In the light of these points, I'd also like the authors to add the word "much" to the following sentence in the abstract: "The re-analysis of these data using the more reliable doubly-ranked stratification method still indicated a potential J-shaped relationship, but with less certainty as there was less precision in estimates at the lower end of the BMI distribution." -> "The re-analysis of these data using the more reliable doubly-ranked stratification method still indicated a potential J-shaped relationship, but with much less certainty as there was less precision in estimates at the lower end of the BMI distribution." Thank you for the explanation of the relationship between the LACE estimates and the smoothed (integral) curves. One minor point is that in Supplementary Figure 3 the LACE estimates are not negative, but below one for low BMI values vs above one on average for higher BMI values. Therefore, the sentence "here, we plot the stratum-specific Mendelian randomization estimates, which reflect the slope of the curves in the main text figures." needs to be corrected (the log of these values reflect the slope in the main figures). But apart from that the author's explanation is correct and sorry for the misunderstanding.
--	---

VERSION 2 – AUTHOR RESPONSE

Reviewer: 1

Comments to the Author:

A0R. Thank you for responding to all comments and adding greater clarity to the manuscript. All comments have been suitably addressed and I have no further comments except for a few minor formatting errors to correct:

> We thank the reviewer for her positive opinion of our submission, and her close reading of our re-submission. <

A1R. Introduction first paragraph – missing units kg/m² for BMI

> We have added units as suggested. <

A2R. Page 10 of revised manuscript: "There was still some indication of a J-shaped curve" – should be "There was..."

> We have corrected this sentence as suggested. <

A3R. In Figures 3 and 4 and Supplementary Figure 5, the x-axis of panel D has much fewer labels (only at BMI of 20, 30, 40), whereas the other panels in the figures use step points of 5 – it would be useful to add a few more labels/make it consistent e.g. 20, 25, 30, 35, 40, 45.

> We have made the x-axis labels consistent within each main figure as requested. We have not made the x-axis labels consistent in Supplementary Figure 5 as the scale of the x-axis depends on the number of strata. <

A4R. I'm not sure if it is just something wrong with the way the PDF was rendered but Supplementary Figure 3 seems to have a superimposed label for panel D (with B and UK Biobank doubled under the text)

> We have found the reason for this superimposed label, and have corrected this. <

A5R. Discussion: “Our analysis has several strengths, but also limitations” – might not need to say “but also limitations” here since the next paragraph says “However, there are important limitations”

> We have chosen to retain this phrasing. While it is slightly repetitive, it is important to point out that all analyses have limitations. <

Reviewer: 2

Comments to the Author:

B0R. I thank the authors for reassuringly addressing most of my points. I still stand by my opinion that the evidence shown here for a J-shaped relationship is largely insufficient and the main conclusion of the paper should rather be that a better method (and follow-up analysis) invalidates the original paper and more data is needed to reassuringly confirm a J-shaped relationship (even if there is one it is very mild).

Having considered all pros and cons, I think it is still better to publish a new version of the retracted paper, even if it represents little evidence for a J-shaped relationship, but at least the analysis is done very carefully.

> We appreciate the reviewer's comments, in particular his support for re-publication using the updated method. We take the reviewer's point that evidence for a J-shaped relationship is limited and somewhat equivocal, and seek to find the best way to express this to both detailed and casual readers. <

I only have a few remaining outstanding comments I would like to insist on:

B1R. In order to transparently present the pros and cons for the J-shaped relationship, I would like the authors briefly repeat all arguments that are not in support for a J-shaped relationship in one place in the Discussion. Namely,

1. When using 200 bins, the HUNT study shows no evidence for a J-shaped relationship.
2. The best fitting non-linear model is not significantly better than the linear causal model. (While a smoothed J-shaped curve looks nice, the truth is that a linear function would fit the data just as well.)

3. Using the extended PRS in the UK Biobank leads to a perfectly linear trend for BMI >21, which represents >95% of the UKB population. Thus evidence for J-shape is even weaker than in the old PRS. This is a substantial shift compared to the minimum value at 25 in the original analysis (~35th percentile vs 5th).

4. Even with 100 bins, in HUNT, the slope in null up to a BMI of 28.

> We appreciate the reviewer's desire for the evidence to be presented in the clearest way possible. We have some reservations about whether it is better to set an analysis plan before looking at the data and stick to that analysis plan, versus performing several analyses using different settings and highlighting the most extreme results (which will inevitably lead to selective reporting). The latter approach seems more open to abuse, as if you run an analysis enough times with subtly different settings, you will eventually get a set of results that you want. Hence while we appreciate the sentiment of the reviewer's comment, we would want to emphasize results from the primary analysis versus drawing attention to specific features of one of two supplementary analyses. We note in passing that the results in Supplementary Figure 4 (log-transformed exposure values) show stronger support for a J-shaped relationship than main analyses – we do not highlight these results in the Discussion section (but if we re-iterated in detail all the points stated by the reviewer, it would be unbalanced not to mention this analysis).

> One point that we would challenge is the relevance of the non-significant p-value for non-linearity. First, we would always be cautious about over-interpreting a null significance test. Secondly, the non-linearity test assesses evidence across the whole distribution. A randomized controlled trial may give a null overall estimate even if there is a significant finding in a small subgroup of the population. Similarly, there may be genuine evidence for an upturn at the lower end of the BMI distribution even if the overall curve is not statistically distinct from linear. Hence while we are happy to re-state that p-values for non-linearity were non-significant, this does not preclude the possibility of a J-shape with upturned risk for a minority of participants.

> We have revised this paragraph to read (additional text is underlined): "In contrast, the doubly-ranked method provided little evidence for non-linearity; formal statistical tests for non-linearity did not reject the null hypothesis of linearity in any analysis. While there was some indication of a J-shaped curve in the main analyses, estimates at the lower end of the BMI distribution were imprecise. One reason for this is that genetic associations with BMI were weaker at low levels of BMI in the doubly-ranked method, and so stratum-specific Mendelian randomization estimates are less precise in these strata. In some supplementary analyses using different numbers of strata, evidence for a J-shaped relationship in the HUNT dataset was absent, and the J-shape was less pronounced when using an extended genetic score in UK Biobank."

> We believe that this text provides a fair summary of the evidence. If the journal editors have further suggestions, then we would be happy to consider these. <

B2R. In the light of these points, I'd also like the authors to add the word "much" to the following sentence in the abstract:

"The re-analysis of these data using the more reliable doubly-ranked stratification method still indicated a potential J-shaped relationship, but with less certainty as there was less precision in estimates at the lower end of the BMI distribution." -> "The re-analysis of these data using the more reliable doubly-ranked stratification method still indicated a potential J-shaped relationship, but with much less certainty as there was less precision in estimates at the lower end of the BMI distribution."

> We have added the word "much" as requested. <

B3R. Thank you for the explanation of the relationship between the LACE estimates and the smoothed (integral) curves. One minor point is that in Supplementary Figure 3 the LACE estimates are not negative, but below one for low BMI values vs above one on average for higher BMI values. Therefore, the sentence “here, we plot the stratum-specific Mendelian randomization estimates, which reflect the slope of the curves in the main text figures.” needs to be corrected (the log of these values reflect the slope in the main figures). But apart from that the author’s explanation is correct and sorry for the misunderstanding.

> We thank the reviewer for acknowledging the misunderstanding. We accept the reviewer’s point that these estimates are hazard ratios, whereas the slopes in Figures 2, 3, and 4 are log hazard ratios, and have corrected the text in the caption of Supplementary Figure 3 to acknowledge this point as requested: “These plots differ from the presentation in the main text Figures: here, we plot the stratum-specific Mendelian randomization estimates, which (when log-transformed) reflect the slope of the curves in the main text figures.”. <